# A Novel Approach by Spark Plasma Sintering to the Improvement of Mechanical Properties of Titanium Carbonitride-Reinforced Alumina Ceramics

**DOI:** 10.3390/molecules26051344

**Published:** 2021-03-03

**Authors:** Magdalena Szutkowska, Marcin Podsiadło, Tomasz Sadowski, Paweł Figiel, Marek Boniecki, Daniel Pietras, Tomasz Polczyk

**Affiliations:** 1Łukasiewicz Research Network-Cracow Institute of Technology, Centre of Advance Manufacturing Technology, 73 Zakopianska St., 30-418 Cracow, Poland; magdalena.szutkowska@kit.lukasiewicz.gov.pl (M.S.); marcin.podsiadlo@kit.lukasiewicz.gov.pl (M.P.); pawel.figiel@kit.lukasiewicz.gov.pl (P.F.); tomasz.polczyk@kit.lukasiewicz.gov.pl (T.P.); 2Faculty of Civil Engineering and Architecture, Lublin University of Technology, 38d Nadbystrzycka St., 20-618 Lublin, Poland; d.pietras@pollub.pl; 3Łukasiewicz Research Network–Institute of Microelectronics and Photonics, 32/34 al. Lotników, 02-668 Warsaw, Poland; Marek.Boniecki@imif.lukasiewicz.gov.pl

**Keywords:** Al_2_O_3_-ZrO_2_ ceramics, Ti(C,N), SPS, pressureless sintering, specific wear rate, K_IC_, X-ray analysis

## Abstract

Ti(C,N)-reinforced alumina-zirconia composites with different ratios of C to N in titanium carbonitride solid solutions, such as Ti(C_0.3_,N_0.7_) (C:N = 30:70) and Ti(C_0.5_,N_0.5_) (C:N = 50:50), were tested to improve their mechanical properties. Spark plasma sintering (SPS) with temperatures ranging from 1600 °C to 1675 °C and pressureless sintering (PS) with a higher temperature of 1720 °C were used to compare results. The following mechanical and physical properties were determined: Vickers hardness, Young’s modulus, apparent density, wear resistance, and fracture toughness. A composite with the addition of Ti(C_0.5_,N_0.5_)n nanopowder exhibited the highest Vickers hardness of over 19.0 GPa, and its fracture toughness was at 5.0 Mpa·m^1/2^. A composite with the Ti(C_0.3_,N_0.7_) phase was found to have lower values of Vickers hardness (by about 10%), friction coefficient, and specific wear rate of disc (Ws_d_) compared to the composite with the addition of Ti(C_0.5_,N_0.5_). The Vickers hardness values slightly decreased (from 5% to 10%) with increasing sintering temperature. The mechanical properties of the samples sintered using PS were lower than those of the samples that were spark plasma sintered. This research on alumina–zirconia composites with different ratios of C to N in titanium carbonitride solid solution Ti(C,N), sintered using an unconventional SPS method, reveals the effect of C/N ratios on improving mechanical properties of tested composites. X-ray analysis of the phase composition and an observation of the microstructure was carried out.

## 1. Introduction

After functional materials for devices in the broadly accepted field of electronics, the most important groups of ceramic materials are advanced ceramic nitride, silicon carbide, alumina, and zirconia materials. The oldest and most commonly used group of ceramic tool material is alumina, denoted as α-Al_2_O_3_. This modification of alumina has a hexagonal crystal lattice with parameters a = 4.754 Å and c = 12.99 Å, where the ions O^2−^ are placed in a close hexagonal structure, with the cations Al^3+^ occupying two-thirds of the octahedral interstitial positions. Ions with high-value charges and the closest packing density are characterized by a high degree of stability, a high melting point, and a high value for hardness [1]. Pure α-Al_2_O_3_ has a high melting point of 2050 °C, a high hardness value (18–20 Gpa), a transverse rupture strength (TRS) of 300–400 Mpa, a low dielectric constant (9.0–10.1 at room temperature), high thermal conductivity (28–35 Wm^−1^K^−1^ at room temperature), and a thermal expansion coefficient (8 × 10^−6^ K^−1^) [2,3]. Due to these properties, alumina can be used in difficult operating conditions. An increase in temperature on the tool blade during its operation requires the use of materials resistant to high temperatures. Therefore, the material used should be characterized by a high thermal conductivity and a low coefficient of thermal expansion. This allows for the attainment of a high degree of resistance for the cutting blade in conditions of rapid temperature changes. Unfortunately, the intrinsic drawbacks of alumina ceramics, such as low strength, low fracture toughness (3–4 Mpa·m^−1/2^), and low thermal shock resistance, significantly limit their wider use as a tool material. Improving the mechanical properties of tool ceramics using alumina reinforced with carbides, oxides, and nitrides was the main goal of this research. Among the commonly used reinforcing phases, hard particles such as Ti(C,N) are preferred for their high hardness value, low density, high melting point, high Young’s modulus, high wear and corrosion resistance, and high thermal stability [4,5,6]. Zirconium dioxide (ZrO_2_) is often used as a phase to increase the fracture toughness of ceramics due to the martensitic transformation from the monoclinic (m) to the tetragonal (t) phase [7,8]. Due to isomorphism of TiC and TiN, the C atoms in the TiC superlattice can be replaced by N atoms in any proportion. Therefore, in this case a binary solid solution, titanium carbonitride is formed with a wide range of compositions: Ti(C_1−x_N_x_) where 0 ≤ x ≤ 1 [9,10]. Changes in the value of x strongly affect the properties of TiCN. Normally, with decreasing x, the unit cell parameter of TiCN decreases linearly and the hardness is lower, but in this case, in contrast, the toughness was enhanced [11,12]. Titanium carbonitride exhibits improved mechanical properties compared to TiC in view of TiN in solid solutions with TiC and combines a high degree of hardness, good wear resistance with the low friction coefficient of TiC, and the higher value of toughness and chemical resistance of TiN [13,14]. In order to find more effective solutions to improving the mechanical properties of alumina composites, the following methods of sintering were used: pressureless sintering (PS), conventional sintering, two-step microwave sintering, hot pressing (HP), and spark plasma sintering [15,16]. In contrast, spark plasma sintering (SPS) is a widely used process which provides a means for ceramic powder sintering using a rapid technique to achieve the full density of the ceramic composite at lower temperatures and with shorter soaking times. An advantage of the SPS method is a short sintering time, taking only a few minutes [17,18]. The basic configuration of a typical SPS system is presented in Figure 1.

The primary objective of the present study was to improve the mechanical properties of Ti(C,N)-reinforced alumina-zirconia composites with a different ratio of C to N in titanium carbonitride solid solution. As a starting powder, Ti(C_0.3_,N_0.7_) (C:N = 30:70) and Ti(C_0.5_,N_0.5_) (C:N = 50:50) were used. The effect of various sintering techniques such as spark plasma sintering and pressureless sintering (PS) on mechanical properties was studied. Moreover, the study was carried out on alumina-zirconia composites reinforced with Ti(C,N) prepared on the basis of micro and nanoscale trade powders. The effect of the Ti(C_0.5_,N_0.5_) powder of nanoscale size on the mechanical properties of the tested composites compared to that of composites with Ti(C_0.5_,N_0.5_) powder of a microscale size was analysed. The new expectation in modern industries for the machining of materials subject to demanding conditions is related to the high precision, high speed, and high quality of the tool materials used. The cutting tools that operate in severe, and especially under intermittent, cutting conditions must have high hard and wear resistance [19]. One widely used consolidation of alumina-zirconia matrix composites reinforced with titanium carbonitride is pressureless sintering. The high density of these composites is possible to obtain at much higher temperatures than for sintering using SPS. Under certain circumstances the cost benefits of pressureless sintering determine its application, but sintering with pressure remains crucial in ceramics engineering for the fabrication of composites at a much lower temperature [20]. Dense spark plasma sintering from a green compact completed in the short time of 10 min was used in this study. To date, there have been no studies of the mechanical properties of Al_2_O_3_/ZrO_2_/Ti(C,N) composites with different ratios of C to N in titanium carbonitride solid solution. Zirconia-alumina matrix composites reinforced with the addition of Ni, Ti, and Mo as a binder phase were sintered by microwave, repetitious-hot-pressing, and HIP (hot isostatic pressing) and tested [13,14]. There has also not been any analysis of the influence of Ti(C,N) powder addition of a nanoscale size on the properties of alumina-zirconia matrix composites. Micro-nano composites of Al_2_O_3_/Ti(C,N) with the addition of WC, Ni, and Mo were sintered by microwave [15]. The SPS sintering of the nanocomposite Ti(C,N)-based cermet with chromium, tungsten carbides, nickel, and molybdenum were studied. The effects of the C/N ratio, WC/Ti(C,N) ratio, and Co/N ratio in the Ti(C,N) solid solution on the properties and microstructure of Ti(C,N) matrix cermets, sintered in a vacuum furnace, were analysed. Three kinds of Ti(C,N) solid solution with different C/N ratios, Ti(C_0.7_,N_0.3_), Ti(C_0.5_,N_0.5_), and Ti(C_0.4_,N_0.6_), were used to prepare the cermets [21]. The results of the research on the mechanical properties of composites, based on alumina-zirconia matrices with different zirconia polymorphic modifications and addition of TiC and TiN, induced the authors to continue research on this issue [22]. We propose a novel approach to the improvement of mechanical properties of an alumina matrix composite by the addition of 2 wt % ZrO_2_
^(m)^ reinforced with solid solution Ti(C,N). Our research of alumina-zirconia composites with the different ratios of C to N in titanium carbonitride solid solution Ti(C,N) reveals an effect of C/N ratios on improving the mechanical properties of Ti(C,N)-reinforced alumina-zirconia composites. It is shown that the Al_2_O_3_/ZrO_2_/Ti(C,N) composites with Ti(C_0.3_, N_0.7_) exhibited the lowest values of friction coefficient and specific wear rate compared to the composite with the addition of Ti(C_0.5_,N_0.5_). The promising results of this study may allow for new alumina-zirconia matrix composites reinforced with titanium carbonitride and sintered using SPS to meet these requirements and thus to expand the range of tool materials used for precision machining.

## 2. Materials and Methods

### 2.1. Materials

Commercial powders of Al_2_O_3_, ZrO_2_, and titanium carbonitride were used as basic materials. A high purity α-Al_2_O_3_ (99.8%) powder type A16SG (Almatis, Inc. Leetsdale, PA, USA) with an average particle size of less than 0.5 μm was used, and Ti(C,N) powder with a microsize of (3–4 μm) and a different ratio of C:N = 30:70 and C:N = 50:50 was also utilized. Powders of Ti(C_0.3_,N_0.7_) and Ti(C_0.5_,N_0.5_) produced by H.C. Starck, Germany were used (Figure 1a). Additionally, Ti(C_0.5_,N_0.5_)n with a nanoscale (40 nm), which was produced by PlasmaChem GmbH, Germany, was applied as a new component of the powder mixture (Figure 2).

The monoclinic phase of zirconia in microsize ZrO_2_^(m)^ (Fluka, Buchs, Switzerland) with a mean particle size of 0.4 μm was added in the amount of 2 wt %. A small amount of sintering additives, such as MgO, to inhibit grain growth was also used. The phase composition of the tested compounds is presented in Table 1.

The components were mixed for about 30 h in Al_2_O_3_ mills with ZrO_2_ balls, with the plasticizer. Mixed powders, which were uniformly set, after plasticizing and drying, were granulated and placed into the graphite element. The compounds were sintered in a furnace type: HP D5 (FCT Systeme GmbH, Rauenstein, Germany) with argon atmosphere in a single technological stage of spark plasma sintering. The powder was pressed at 90 MPa into a disk with a diameter of 20 mm and about 5.0 mm in thickness and then consolidated under a pressure of 35 MPa at various temperatures ranging from 1600 °C to 1675 °C and at a heating rate of 100 °C/min. Each sample was maintained at a peak temperature for 10 min. The SPS process of alumina-zirconia powders with Ti(C_0.5_,N_0.5_)n is shown in Figure 3. Furthermore, green compact samples with dimensions of 13.5 × 13.5 × 5.5 mm, were uniaxially pressed at 100 MPa and then isostatically cold pressed at 200 MPa and sintered using pressureless sintering (PS).

An HTK8 Gero furnace with an argon atmosphere was used. The green compacts were sintered for 60 min at a temperature of 1720 °C. The heating rate was 100 °C/min. The sintering temperatures were controlled using an optical pyrometer. The samples were plane ground with diamond (MD-Piano120, Struers, Ballerupt, Denmark), fine ground with diamond grains of about 3 μm (MD-Largo, Struers Ballerupt, Denmark), and finally polished.

### 2.2. Characterization Techniques

The physical and mechanical properties of Ti(C,N)-reinforced alumina-zirconia composites were determined using the following measurements: Vickers hardness *HV1*, Young’s modulus *E,* friction coefficient *μ*, (in ball-on-disc tests, using a CETR UMT-2MT universal mechanical tester (Cetr, Inc., Campbell, CA, USA), and apparent density *ρ*. A device, hardness tester (FUTURE-TECH Corp., Kawasaki, Japan), was used to measure Vickers hardness *HV1* at loading force *9.81 N*. Based on an ultrasonic method, Young’s modulus E was calculated [23].

Three-point bending method of SENB (Single-Edge Notched Beam) samples(with dimensions of 2.5 × 4.0 × 30.0 ± 0.1 mm) was used for the measurement of fracture toughness—see Figure 4.

An initial notch of 0.9 mm deep was cut by a diamond saw (thickness: 0.20 mm), and then the notch tip was pre-cracked with a thin diamond saw (thickness: 0.025 mm). The total initial notch length was approximately 1.1 mm. The relationship *K*_IC_ = *f(c)* is given by Equations (1) and (2) [24]:(1)$$K_{IC}=1.5\frac{P_{c}^{}S}{W^{2}B}Yc^{1/2}$$
and
(2)$$Y=\frac{\sqrt{\Pi }}{{\left(1-\beta \right)}^{{}_{2}^{3}}}\left[0.3738\beta +(1-\beta )\sum_{i,j=0}^{4}A_{ij}\beta^{i}{\left(\frac{W}{S}\right)}^{j}\right]$$
where *P_c_* is the critical load, *S* is the support span, *W* is the width, *B* is the sample thickness, *Y* is the geometric function, *c* is the crack length, *β* is *c*/*W*, and *A_ij_* refers to the coefficients given by Fett [25].

The friction coefficient and the specific wear rate of the disk-shaped samples in contact with the Al_2_O_3_ ball were measured at room temperature. In this method, an extensometer was used to measure the friction force. The disk-shaped samples were 20 mm in diameter and about 5.0 mm in height. The following parameters were applied in the test: a ball diameter of 3.2 mm, a loading force of 20 N, a sliding speed of 0.1 m/s, a sliding circle radius of 4 mm, a sliding distance of 2000 m, and 20,000 s for the time of the test. The measurements were made without using lubricant. The results were obtained from three samples according to the ISO 20808:2016 standard [26]. The volume of the removed material was determined from the cross-sectional area of the wear track and its circumference. The specific wear rate was defined [22]. A JEOL JSM-6460LV scanning electron microscope provided with a set of X-ray spectrometers (by Oxford Instruments, Abingdon, England) was used for the microstructural image. XRD measurements were taken using a PANAlytical Empyrean system with Cu Kα1 radiation. The phase compositions of the sintered compacts were identified using the database of the International Centre for Diffraction Data PDF4+2018. The quantitative compositions of the samples were determined using the Rietveld method with the X’Pert Plus program.

## 3. Results and Discussion

### 3.1. Mechanical and Physical Properties

The average values of the selected mechanical and physical properties (Vickers hardness, apparent density, Young’s modulus, fracture toughness) of the tested alumina matrix composites TICN5, TICN5n, and TICN3, sintered using the SPS technique at various temperatures from 1600 °C to 1675 °C, are recorded in Table 2.

The titanium carbonitride-reinforced alumina-zirconia matrix composites sintered through SPS revealed the following: a Vickers hardness ranging from 16.4 ± 0.2 to 19.5 ± 0.2 GPa, a Young’s modulus ranging from 400 ± 5.0 to 410 GPa ± 5.0, a critical stress intensity factor *K*_IC_ (3PB test with SENB samples) ranging from 4.69 ± 0.1 to 5.29 ± 0.3 MPa·m^1/2^, and an apparent density ranging from 4.26 ± 0.01 to 4.30 ± 0.01 g/cm^3^. The composite with the addition of Ti(C_0.5_,N_0.5_)n nanopowder (samples marked as TICN5n) exhibited the highest Vickers hardness value, over 19.0 GPa, and its fracture toughness was 5.0 MPa·m^1/2^. However, the samples marked TICN5 and TICN3 revealed from 5% to 10% lower Vickers hardness values compared to the TICN5n sample. The Vickers hardness values decrease slightly from 5% to 10% with increasing sintering temperature—see Figure 5.

The properties of the two kinds of Ti(C,N) solid solution with different C/N ratios Ti(C_0.3_,N_0.7_) and Ti(C_0.5_,N_0.5_) used to fabricate the Al_2_O_3_/ZrO_2_/Ti(C,N) composites were analysed—see Table 2. It should be noted that slightly higher values (above 5.0 MPa·m^1/2^) of fracture toughness were attained by the TICN3 sample with a high N:C ratio in the Ti(C_0.3_,N_0.7_) solid solution. There is only a 2–7% difference in *K*_IC_ compared to the TICN5 composites with the addition of Ti(C_0.5_,N_0.5_). The average values of Vickers hardness, apparent density, Young’s modulus, and fracture toughness of the tested alumina matrix composites TICN5, TICN5n, and TICN3, sintered using the PS technique at a temperature of 1720 °C, are recorded in Table 3.

The results produced by the alumina-zirconia matrix composites with titanium carbonitride, sintered using the pressureless method, show a Vickers hardness HV1 ranging from 16.8 to 17.6 GPa, a Young’s modulus ranging from 375 to 400 GPa, a fracture toughness *K*_IC_ ranging from 4.32 to 5.10 MPa·m^1/2^, and an apparent density ranging from 4.218 to 4.24 g/cm^3^. The Vickers hardness HV1 and the Young’s modulus of these samples are 10–15% lower when compared to those samples with the same chemical composition but sintered at the lowest temperature (1600 °C) using SPS. In the temperature range of the studies, the SPS sintering temperature has a minor effect on the fracture toughness of the tested samples. The fracture toughness of the TICN5n and TICN3 samples, sintered at a temperature of 1720 °C by PS, revealed lower values when compared to those sintered at the highest temperature (1675 °C) using the SPS method—see Figure 6.

### 3.2. Microstructural Observations

Both transgranular and intergranular fractures occurred during Vickers crack propagation in tested composites (Figure 7). Most intergranular fractures revealed Al_2_O_3_/ZrO_2_/Ti(C,N) composites with Ti(C_0.5_,N_0.5_). Moreover, the presence of crack bridging and crack deflection in these samples had an effect on the enhancement of fracture toughness (Figure 7a).

The microstructures of the fracture surface after the 3PB test in Al_2_O_3_/ZrO_2_/Ti(C,N) composites sintered using SPS at a temperature of 1650 °C is shown in Figure 8. In this case, the fracture surfaces are characterized by intergranular and transgranular fractures. Cracks propagate both along the boundaries and through the grains of Al_2_O_3_ and Ti(C,N). For the most part, the transgranular fractures occur with typical cleavage steps which are related to the bigger grains of Al_2_O_3_ and Ti(C,N) (Figure 8b,c). This occurs when the interfacial bonding strength of the spark plasma-sintered composite is high and particularly when it is even higher than the strength of some grains. Moreover, a transgranular fracture consumes more fracture surface energy, which is beneficial for mechanical properties [15]. Microstructural observations of the tested alumina-zirconia matrix composites reinforced with carbonitrides confirmed their favourable consolidation after sintering.

### 3.3. X-ray Diffraction Analysis

The X-ray diffraction analysis of the tested composites with different ratios of C to N in titanium carbonitride, Ti(C_0.3_N_0.7_), Ti(C_0.5_N_0.5_), and Ti(C_0.5_N_0.5_)n, sintering at 1600 °C and 1675 °C (identified as TICN3.1, TICN3.3, TICN5.1, TICN5.3, TICN5.1_n_, and TICN5.3_n_, respectively), revealed the presence of the following phase: Al_2_O_3_, Ti(C,N), and ZrO_2_ in monoclinic and tetragonal forms (Table 4). The partial polymorphic transformation of monoclinic ZrO_2_^(m)^ into tetragonal ZrO_2_^(t)^ in the tested composites was observed. The tetragonal zirconia phase content ranged from 0.5 to 1.0 wt %. The most intense transformation was observed in the Al_2_O_3_/ZrO_2_/Ti(C_0.5_, N_0.5_) composites. The percentage of Ti(C_0.5_,N_0.5_) solid solution in the tested composites ranged from 23.9 to 30.0 wt %.

An example of the X-ray diffraction analysis of TICN5.1 and TICN5.3 is presented in Figure 9.

The qualitative phase composition of the Al_2_O_3_/Ti(C,N)/ZrO_2_ samples with Ti(C,N) powder in micrometric sizes (TACN) and mixed micrometric and nanometric (TACNn) sizes was the same. However, the sample marked TICN3.3 was found to contain a negligible amount of TiO. No significant changes were observed to the percentages of alumina, Ti(C_0.5_, N_0.5_) solid solution, and ZrO_2_. The exception is the TICN5.3 sample, which was sintered at the higher temperature of 1675 °C, for which the ZrO_2_ content decreased from 2% to 1.2%. The ratio of ZrO_2_^(m)^ to ZrO_2_^(t)^ changes in the tested composites. The highest 3/1 ratio of ZrO_2_^(m)^ to ZrO_2_^(t)^ is present in samples marked TICN5.1n and TICN5.3n.

### 3.4. Tribological Properties

The friction coefficient of the tested composites sintered using SPS was in the range of 0.39 to 0.63—see Figure 10.

The sintering temperature used in this test had an effect on the friction coefficient. The results show that with an increase of the sintering temperature from 1600 °C to 1675 °C, the friction coefficient decreased about 20% for composites TICN5 and TICN3, whereas the TICN5n composites reveals an inverse correlation. The highest value of friction coefficient (0.63) was exhibited by the TICN5 composite at a temperature of 1600 °C. The friction coefficient of sample TICN5n, with the addition of Ti(C_0.5_,N_0.5_)n powder with a nanometric size to the alumina-zirconia matrix, tested at 1600 °C, was lower by about 25% compared to sample TICN5 with the Ti(C_0.5_,N_0.5_) powder of a micrometric size. The lowest value of friction coefficient (0.39) was attained by sample TICN3.3 with the Ti(C_0.3_, N_0.7_). The specific wear rate of the disc (Ws_d_) tested at the same sintering temperature of 1600 °C was 0.26 × 10^−6^ and 0.30 × 10^−6^ for TICN5.1 and TICN5.1_n_ respectively. The TICN3.2 sample had the lowest (0.2 × 10^−6^) specific wear rate for the disc (Ws_d_). The microstructure of the wear track for the TICN5.1 sample differed from that of TICN5.1n—see Figure 11. The grain fracture of the TICN5.1n sample may create a wear debris and an increase in wear rate.

Mechanical properties of Ti(C,N)-reinforced alumina-zirconia ceramics (with various C/N ratios) sintered using SPS method have not been discussed in the literature. Therefore, it is difficult to compare the results obtained in this study with other results obtained in similar conditions. The effect of the WC/TiCN ratio, Co/Ni ratio, and C/N ratio on the properties and microstructures of Ti(C,N) matrix cermets, sintered in a vacuum furnace, was investigated. A Ti(C,N) solid solution with different ratios of C to N (70/30, 50/50, and 40/60) in Ti(C,N) matrix cermets was analysed. It was found that the cermet with a Ti(C_0.5_, N_0.5_) solid solution had a higher hardness HRA and transverse rupture strength (TRS) than the others [21]. In most TiCN-based cermets, the binder phase is mainly composed of metals, and the solid solution hardening of the binder phase plays an important role in the application of cermets [6]. While increasing interest in Ti(C,N)-based cermets as a tool material has been observed from the early 1970s, improvements in the mechanical properties of these materials through the application of novel techniques of consolidation were widely discussed in the last decade [27,28,29,30,31]. Therefore, Ti(C,N)-based cermets with binder metals (Mo, Ni, and Co) and different secondary carbides (WC, NbC, TaC, HfC, VC, Cr_3_C_2_, and Mo_2_C) were manufactured using the vacuum hot-pressing process, the mechanically induced self-sustaining reaction process, microwave sintering, spark plasma sintering, and sintering in an induction furnace [21,27,28,29,30,31,32]. Alumina matrix composites reinforced with 30 wt % Ti(C_0.7_, N_0.3_) were densified by a two-stage gas pressure sintering technique [33]. A high hardness value of over 19.0 GPa for the tested composites resulted from solid state sintering at a high temperature of over 1800 °C. Indentation fracture toughness obtained at a load of 49 N was 5.85 MPa·m^1/2^. The microstructure and properties of Al_2_O_3_–ZrO_2_(Y_2_O_3_) ceramics with a ratio of Al_2_O_3_ to ZrO_2_(Y_2_O_3_) of 50:50 and 70:30 respectively, and an addition of 20 vol % titanium carbonitride after hot pressing at 35 MPa and temperatures of 1500 °C and 1550 °C, were analysed [34]. The 70Al_2_O_3_–30ZrO_2_(Y_2_O_3_) composites with a 20 vol % of Ti(C,N) exhibited high values of indentation fracture toughness (8–9 MPa·m^1/2^), but their Vickers hardness was only 17.5 HV_10_. An Al_2_O_3_-based functionally graded ceramic tool material reinforced with TiCN microparticles, a small amount of Ni and Mo, and nano-Al_2_O_3_ powders was produced using a hot-pressing technique [35]. The excellent mechanical properties were received for the cermets with Al_2_O_3_ nanoparticles increasing from 10 vol % in the surface to 20 vol % in the core, with the flexural strength, fracture toughness, and Vickers hardness being 1073 MPa, 5.99 MPa·m^1/2^, and 21.78 GPa, respectively.

## 4. Conclusions

Ti(C,N)-reinforced alumina-zirconia composites with different ratios of C to N in titanium carbonitride solid solution, such as Ti(C_0.3_,N_0.7_) (C:N = 30:70) and Ti(C_0.5_,N_0.5_) (C:N = 50:50), were tested to improve their mechanical properties. The effect of the Ti(C_0.5_,N_0.5_)n powder of nanoscale size on the mechanical properties of the tested composites compared to composites with Ti(C_0.5_,N_0.5_) powder of microscale size was analysed.The composite with the addition of Ti(C_0.5_,N_0.5_) powder of nanoscale size exhibited the highest Vickers hardness (over 19.0 GPa), and its fracture toughness was 5.0 MPa·m^1/2^.It was found that slightly higher values (above 5.0 MPa·m^1/2^) of fracture toughness were achieved by a composite with C:N = 30:70 ratio in the Ti(C_0.3_,N_0.7_) solid solution. There was only a 2–7% difference in *K*_IC_ compared to composites with the addition of Ti(C_0.5_,N_0.5_).The Vickers hardness values decreased slightly with increasing sintering temperature. The difference in the Vickers hardness values between the tested composites measured at the lowest (1600 °C) and the highest (1675 °C) temperature ranged from 5% to 10%.The fracture toughness of the composites sintered at a temperature of 1720 °C by PS revealed lower values when compared to those sintered at the highest temperature (1675 °C) using the SPS method.The phases of Al_2_O_3_, Ti(C,N), and ZrO_2_ in the monoclinic and tetragonal form were revealed during X-ray diffraction analysis of the tested composites. The partial polymorphic transformation of monoclinic ZrO_2_^(m)^ into tetragonal ZrO_2_^(t)^ in the tested composites was observed. The most intense transformation was observed in Al_2_O_3_/ZrO_2_/Ti(C_0.5_,N_0.5_) composites. The percentage of Ti(C_0.5_,N_0.5_) solid solution in the tested composites was in the range of 23.9 to 30.0 wt %.The sintering temperature used in this test had an effect on the friction coefficient. The highest value of the friction coefficient (0.63) was exhibited in a composite with Ti(C_0.5_, N_0.5_) powder of micrometric size sintered by SPS at a temperature of 1600 °C. The lowest friction coefficient (0.39) and specific wear rate of the disk (0.2 × 10^−6^) was attained by the composite with the addition of Ti(C_0.3_,N_0.7_) to the alumina-zirconia matrix.The experimental testing of the considered composites will be extended to uniaxial tension or compression loading similar to [36,37,38,39,40,41]. The corresponding numerical models will be elaborated for a description of the quasi-static and dynamic response of the analysed Al_2_O_3_/ZrO_2_/Ti(C,N) composites. Moreover, brittle-damage and shear-damage models and surface-based cohesive behaviour will be used to account for decohesion process of composites at interfaces [40,41,42,43,44,45,46,47,48].

## Figures and Tables

**Figure 1 molecules-26-01344-f001:**
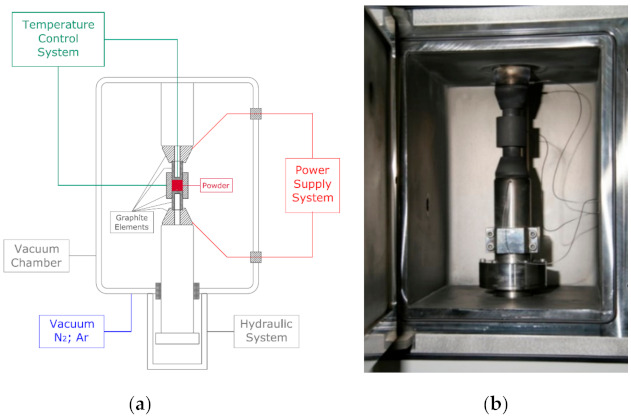
Spark plasma sintering (SPS) system: (**a**) basic configuration and (**b**) vacuum and gas chamber.

**Figure 2 molecules-26-01344-f002:**
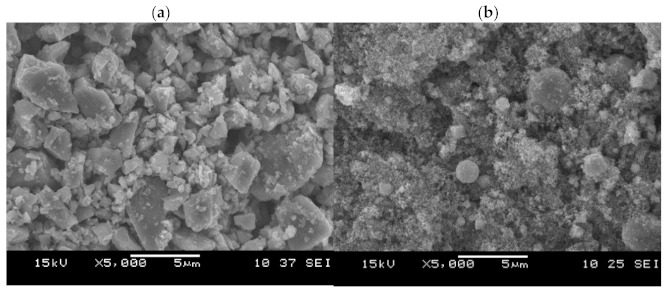
SEM image of commercially available powders: (**a**) Ti(C_0.5_,N_0.5_); (**b**) Ti(C_0.5_,N_0.5_)n.

**Figure 3 molecules-26-01344-f003:**
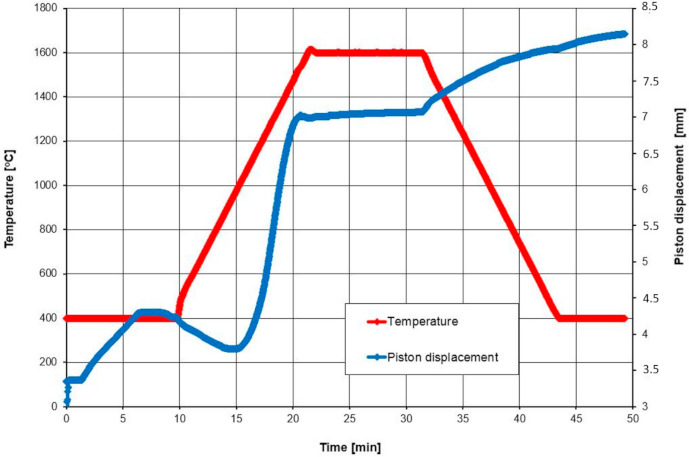
Sintering curve of alumina-zirconia powder with Ti(C_0.5_,N_0.5_)_n_.

**Figure 4 molecules-26-01344-f004:**
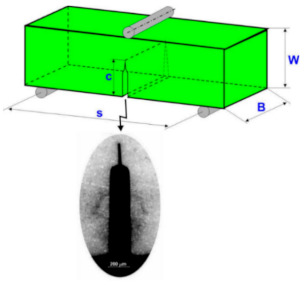
Specimen for the measurement of fracture toughness (*K*_IC_).

**Figure 5 molecules-26-01344-f005:**
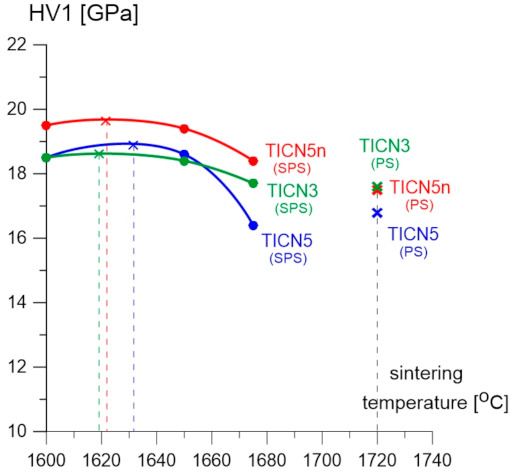
The relationship between Vickers hardness and the temperature of the samples sintered using SPS and PS.

**Figure 6 molecules-26-01344-f006:**
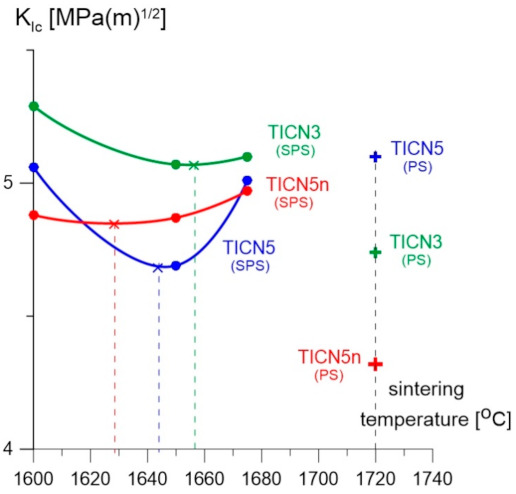
The relationship of *K*_IC_ and temperature of the samples sintered using SPS and pressureless sintering (PS).

**Figure 7 molecules-26-01344-f007:**
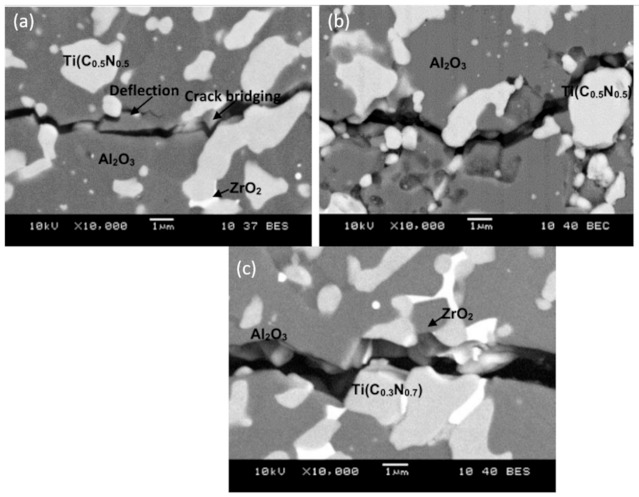
SEM micrograph of the Vickers cracks in Al_2_O_3_/ZrO_2_/Ti(C,N) specimens sintered using SPS at a temperature of 1675 °C: (**a**) TICN5.3; (**b**) TICN5.3n; (**c**) TICN3.3.

**Figure 8 molecules-26-01344-f008:**
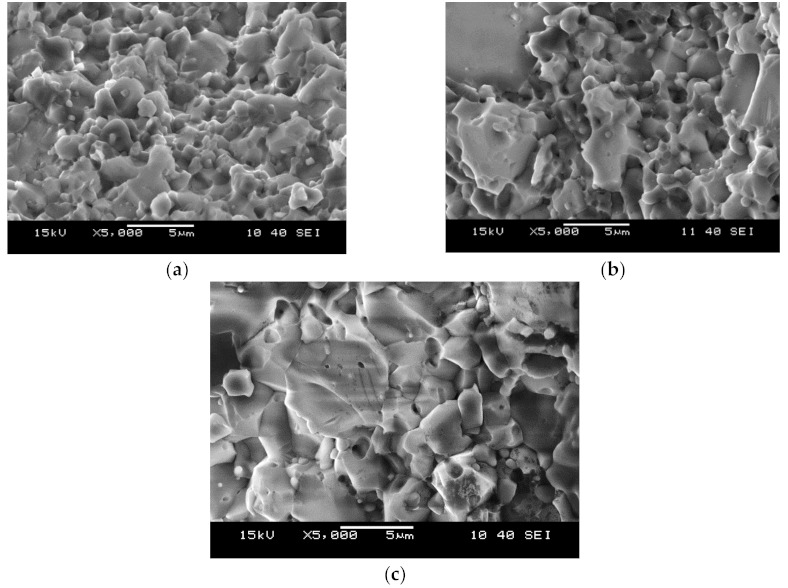
SEM morphologies of the fracture surface of the tested composites (Single-Edge Notched Beam (SENB) sample) sintered at a temperature of 1650 °C: (**a**) TICN5.3; (**b**) TICN5.3n; (**c**) TICN3.3.

**Figure 9 molecules-26-01344-f009:**
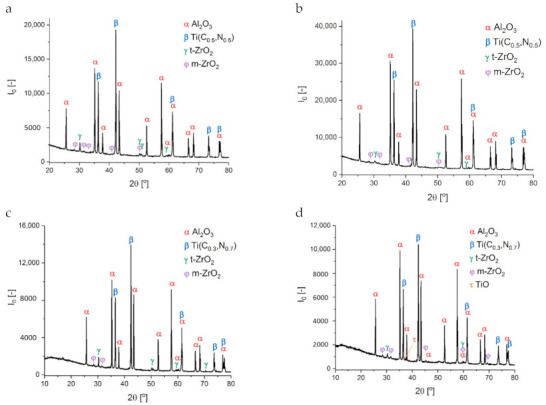
X-ray diffraction analysis of the tested composites, which are identified as (**a**) TICN5.1, (**b**) TICN5.3, (**c**) TICN3.1, and (**d**) TICN3.3.

**Figure 10 molecules-26-01344-f010:**
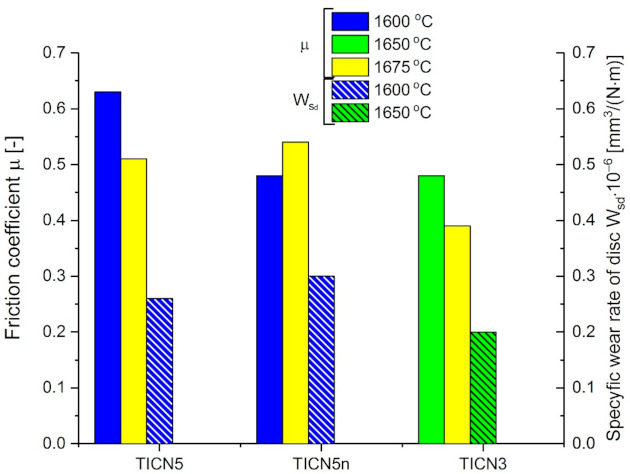
Mean values of the friction coefficient (*μ*) and the disc wear rate (*W*s_d_) of the Al_2_O_3_/Ti(C,N)/ZrO_2_ composites.

**Figure 11 molecules-26-01344-f011:**
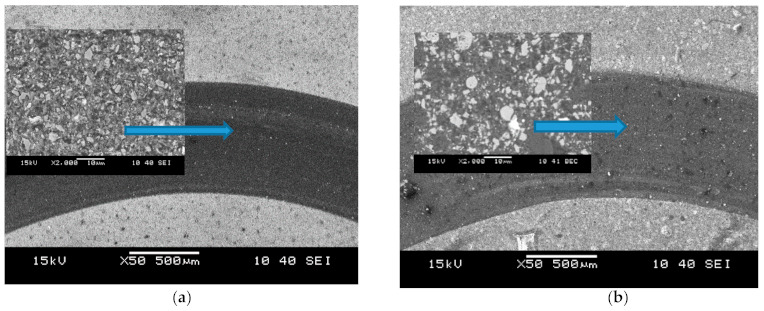
Microstructure of the wear track for composites: (**a**) TICN5.1; (**b**) TICN5.1n.

**Table 1 molecules-26-01344-t001:** Composition of compounds.

Compounds	Components, wt %				
	Al_2_O_3_ + MgO	ZrO_2_^(m)^	Ti(C_0.3_,N_0.7_)	Ti(C_0.5_,N_0.5_)	Ti(C_0.5_,N_0.5_)n
TICN3	68	2	30	-	-
TICN5	68	2	-	30	-
TICN5n	68	2	-	20	10

**Table 2 molecules-26-01344-t002:** Selected mechanical and physical properties of the tested composites sintered using SPS.

Samples	Sintering Temperature (°C)	Vickers Hardness HV1(GPa)	Apparent Density ρ (g/cm^3^)	Young’s Modulus E (GPa)	Fracture Toughness K*_IC_* (MPa·m^1/2^)
TICN5.1	1600	18.5 ± 0.2	4.27 ± 0.01	400 ± 5.0	5.06 ± 0.2
TICN5.2	1650	18.6 ± 0.3	4.27 ± 0.01	410 ± 6.0	4.69 ± 0.1
TICN5.3	1675	16.4 ± 0.2	4.26 ± 0.01	400 ± 5.0	5.01 ± 0.2
TICN5.1n	1600	19.5 ± 0.2	4.28 ± 0.01	400 ± 5.0	4.88 ± 0.1
TICN5.2n	1650	19.4 ± 0.2	4.29 ± 0.01	405 ± 4.0	4.87 ± 0.1
TICN5.3n	1675	18.4 ± 0.3	4.27 ± 0.01	400 ± 5.0	4.97 ± 0.2
TICN3.1	1600	18.5 ± 0.3	4.30 ± 0.01	410 ± 5.0	5.29 ± 0.3
TICN3.2	1650	18.4 ± 0.2	4.29 ± 0.01	400 ± 5.0	5.07 ± 0.2
TICN3.3	1675	17.7 ± 0.2	4.30 ± 0.01	400 ± 5.0	5.10 ± 0.2

**Table 3 molecules-26-01344-t003:** Vickers hardness, apparent density, Young’s modulus, and fracture toughness of the tested composites sintered by PS.

Samples	Sintering Temperature (°C)	Vickers Hardness HV1 (GPa)	Apparent Density ρ (g/cm^3^)	Young’s Modulus E (GPa)	Fracture Toughness *K*_IC_ (MPa·m^1/2^)
TICN5.5	1720	16.8 ± 0.2	4.21 ± 0.01	390 ± 4.0	5.10 ± 0.2
TICN5.5n	1720	17.5 ± 0.2	4.21 ± 0.01	375 ± 4.0	4.32 ± 0.1
TICN3.5	1720	17.6 ± 0.2	4.24 ± 0.01	400 ± 5.0	4.74 ± 0.2

**Table 4 molecules-26-01344-t004:** X-ray diffraction analysis of the tested composites sintered using SPS.

	Content (wt %)				
Composite	Al_2_O_3_	Ti(C_0.5_,N_0.5_)	Ti(C_0.3_,N_0.7_)	ZrO_2_^(m)^	ZrO_2_^(t)^
TICN5.1	68.8	29.3	-	0.9	1.0
TICN5.3	68.8	30.0	-	0.6	0.6
TICN5.1n	70.0	28.0	-	1.5	0.5
TICN5.3n	69.0	29.0	-	1.5	0.5
TICN3.1	74.0	-	23.9	1.3	0.6
TICN3.3 *	68.7	-	29.2	1.2	0.8

* TiO is of 0.2 wt %.

## Data Availability

Not applicable.

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
