# Peer review of "A Novel Approach by Spark Plasma Sintering to the Improvement of Mechanical Properties of Titanium Carbonitride-Reinforced Alumina Ceramics"

_molecules, 2021, doi:10.3390/molecules26051344_

Round 1
Reviewer 1 Report
The current manuscript presents an experimental work about the reinforcement of alumina-zirconia composites by adding Ti(C,N). Mechanical properties of several compositions have been evaluated after spark plasma sintering. To compare, pressureless sintered samples were also produced and investigated.
I DO NOT recommend this manuscript for publication. Lines 116 to 123, there are comments addressed to the coauthors in the manuscript written in Poland, which is very disrespectful to the reviewers. It means this manuscript is under reviewing process by the author and coauthors and could not be submitted to any kind of Journal. Please, before submitting a manuscript, make sure it is ready to be evaluated.
Although it has been stated as a novel approach (line 1), it is not clear in the abstract what is the novelty.
It has been stated in the second phrase of the abstract that the SPS sintering temperature was 1600 to 1675 C. However, by the end of the abstract, the authors mentioned PS sintering temperature of 1720 C. What do the authors mean with PS? There is no connection between SPS e PS sintering temperatures at this point.
In the Introduction section, there are 2 font sizes.
Conclusion is a compilation of the results.
Author Response
Comments and Suggestions for Authors
The current manuscript presents an experimental work about the reinforcement of alumina-zirconia composites by adding Ti(C,N). Mechanical properties of several compositions have been evaluated after spark plasma sintering. To compare, pressureless sintered samples were also produced and investigated.
I DO NOT recommend this manuscript for publication. Lines 116 to 123, there are comments addressed to the coauthors in the manuscript written in Poland, which is very disrespectful to the reviewers. It means this manuscript is under reviewing process by the author and coauthors and could not be submitted to any kind of Journal. Please, before submitting a manuscript, make sure it is ready to be evaluated.
Re: We are very sorry for this polish text. All Polish text was removed.
Although it has been stated as a novel approach (line 1), it is not clear in the abstract what is the novelty.
It has been stated in the second phrase of the abstract that the SPS sintering temperature was 1600 to 1675 C. However, by the end of the abstract, the authors mentioned PS sintering temperature of 1720 C. What do the authors mean with PS? There is no connection between SPS e PS sintering temperatures at this point.
Re: The Abstract was modified. To the best of our knowledge, there have been no studies of mechanical properties improvement of alumina matrix composites reinforced with different ratios of C:N in Ti(C,N) sintered using unconventional SPS method. Therefore, novelty relies on revealing an effect of C:N ratios in Ti(C,N) on the improving mechanical properties of tested composites using SPS technique.
The PS sintering method of Al2O3/ZrO2/Ti(C,N) composites requires using of high temperatures above 1700°C. The lower temperatures does not allow to obtain a high-density of material. An advantage of the SPS sintering technique is possibility to obtain better mechanical properties at much lower sintering temperatures than those required for PS method.
In the Introduction section, there are 2 font sizes.
Re: Font sizes was changed
Conclusion is a compilation of the results.
Re: In conclusions we selected all important obtained results.

Reviewer 2 Report
This work discussed on improving the mechanical properties of Ti(C,N) reinforced alumina-zirconia composites with a different ratio of C:N in titanium carbonitride solid solution. The manuscript contains a novel approach by spark plasma sintering to lowering the sintering temperature of the Al2O3 ceramic material and improving the mechanical properties of alumina-zirconia composites. I think this work is valuable. Thus I recommend it to be published in molecules with the following several minor problems that need to be modified before publishing.
- On page 4,“A small amount of sintering additives, such as MgO, to inhibit grain growth was also used.”Here, what’s the amount of MgO additive? Please give the content of MgO additive in the table 1.
- Some errorsor non-englishoccurred on page 3, line 116-123.
- The diffraction peaks of(101)tand (111)c overlapped with each other because the lattice parameters of the cubic and tetragonal phases ZrO2 were nearly the same. How does the author determine that the zirconium oxide in the composite material is tetragonal? And how to determine the content of ZrO2(m) and ZrO2(t) based on the data of XRD? Please further discuss it.
Author Response
Comments and Suggestions for Authors
This work discussed on improving the mechanical properties of Ti(C,N) reinforced alumina-zirconia composites with a different ratio of C:N in titanium carbonitride solid solution. The manuscript contains a novel approach by spark plasma sintering to lowering the sintering temperature of the Al2O3 ceramic material and improving the mechanical properties of alumina-zirconia composites. I think this work is valuable. Thus I recommend it to be published in molecules with the following several minor problems that need to be modified before publishing.
- On page 4,“A small amount of sintering additives, such as MgO, to inhibit grain growth was also used.”Here, what’s the amount of MgO additive? Please give the content of MgO additive in the table 1.
Re: Content of MgO was included in the Table 1
Some errorsor non-englishoccurred on page 3, line 116-123.
Re: We are very sorry for this Polish text. All Polish text was removed.
- The diffraction peaks of(101)tand (111)c overlapped with each other because the lattice parameters of the cubic and tetragonal phases ZrO2 were nearly the same. How does the author determine that the zirconium oxide in the composite material is tetragonal? And how to determine the content of ZrO2(m) and ZrO2(t) based on the data of XRD? Please further discuss it.
Re:
It can be seen that the signals for the tetragonal and cubic phases overlap. However, for an angle of about 50°, a double signal is observed which can only be assigned to ZrO2 (t) - planes (112) and (200). At this point, the ZrO2 (m) phase has a signal coming from only one plane (200).
The positions of the signals are distinguishable and overlap on the diffraction patterns. The content of the phases was determined by the Rietveld method, which allowed to estimate the content with an uncertainty of 0.2-0.3%.

Reviewer 3 Report
The topic is interesting and the manuscript could be considered for publication after minor revision.
- EDS measurements of the wear track can be add to characterize the composition in wear track
- XRD spectra in Fig.9 show an amorphous contribute . Please describe ?
- Line 109-112 and 116-123: sentences are written in Polish.
- Check the format: text size in the introduction , and table 1 and 2.
Author Response
Comments and Suggestions for Authors
The topic is interesting and the manuscript could be considered for publication after minor revision.
- EDS measurements of the wear track can be add to characterize the composition in wear track
Re:
The measurement of the friction coefficient and the specific wear rate of the disk-shaped samples in contact with the Al2O3 ball was used ( point 2.2). EDS measurement of the wear track did not change the composition in wear track in comparison to the surface before the wear test. Additionally, a mapping of elements on this surface did not show any changes. Therefore, we have not included these results.
- XRD spectra in Fig.9 show an amorphous contribute . Please describe ?
An amorphous phase, as shown in XRD spectra in Fig. 9 is related with background elevation for lower angles when the FDS system is used for presentation of the results. The phase composition of the tested materials does not indicate the presence of an amorphous phase.
Re:
The phase composition of the tested materials does not indicate the presence of an amorphous phase. An amorphous phase, as shown in XRD spectra in Fig. 9 is related with background elevation for lower angles when the FDS system is used for presentation of the results. In FDS mode, a larger area is illuminated when working at low angles of the goniometer, hence the background level is raised.
- Line 109-112 and 116-123: sentences are written in Polish.
Re: We are very sorry for this Polish text. All Polish text was removed.
- Check the format: text size in the introduction , and table 1 and 2.
Re: Format of the text size has been changed
Round 2
Reviewer 1 Report
After revision, I would accept this format for publication.